Cover cropping can be a stronger determinant than host crop identity for arbuscular mycorrhizal fungal communities colonizing maize and soybean

http://orcid.org/0000-0002-9314-6994 Higo Masao higo.masao@nihon-u.ac.jp
Tatewaki Yuya
Gunji Kento
Kaseda Akari
Isobe Katsunori
Department of Agricultural Bioscience, College of Bioresource Sciences, Nihon University , Fujisawa, Kanagawa , Japan
Landa Blanca
Electronic publication date: 2019 Feb 8
Publication date: 2019
Volume: 7
Electronic Location ID: e6403
Received 2018 Aug 24; Accepted 2019 Jan 7
Copyright: © 2019 Higo et al.
Copyright year: 2019
Copyright holder: Higo et al.
License: This is an open access article distributed under the terms of the Creative Commons Attribution License, which permits unrestricted use, distribution, reproduction and adaptation in any medium and for any purpose provided that it is properly attributed. For attribution, the original author(s), title, publication source (PeerJ) and either DOI or URL of the article must be cited.
License URL: https://creativecommons.org/licenses/by/4.0/

Keywords: Amplicon sequencing, Arbuscular mycorrhizal fungi, Cover cropping, Glycine max (L.) Merr., Host identity, Illumina Miseq platform, Zea mays L.

Funding: Nihon University College of Bioresource Sciences Research Grant 17C-001 This work was supported by a grant from the Nihon University College of Bioresource Sciences Research Grant (17C-001). The funders had no role in study design, data collection and analysis, decision to publish, or preparation of the manuscript.

==============================
Background

Understanding the role of communities of arbuscular mycorrhizal fungi (AMF) in agricultural systems is imperative for enhancing crop production. The key variables influencing change in AMF communities are the type of cover crop species or the type of subsequent host crop species. However, how maize and soybean performance is related to the diversity of AMF communities in cover cropping systems remains unclear. We therefore investigated which cover cropping or host identity is the most important factor in shaping AMF community structure in subsequent crop roots using an Illumina Miseq platform amplicon sequencing.

Methods

In this study, we established three cover crop systems (Italian ryegrass, hairy vetch, and brown mustard) or bare fallow prior to planting maize and soybean as cash crops. After cover cropping, we divided the cover crop experimental plots into two subsequent crop plots (maize and soybean) to understand which cover cropping or host crop identity is an important factor for determining the AMF communities and diversity both in maize and soybeans.

Results

We found that most of the operational taxonomic units (OTUs) in root samples were common in both maize and soybean, and the proportion of common generalists in this experiment for maize and soybean roots was 79.5% according to the multinomial species classification method (CLAM test). The proportion of OTUs specifically detected in only maize and soybean was 9.6% and 10.8%, respectively. Additionally, the cover cropping noticeably altered the AMF community structure in the maize and soybean roots. However, the differentiation of AMF communities between maize and soybean was not significantly different.

Discussion

Our results suggest cover cropping prior to planting maize and soybean may be a strong factor for shaping AMF community structure in subsequent maize and soybean roots rather than two host crop identities. Additionally, we could not determine the suitable rotational combination for cover crops and subsequent maize and soybean crops to improve the diversity of the AMF communities in their roots. However, our findings may have implications for understanding suitable rotational combinations between cover crops and subsequent cash crops and further research should investigate in-depth the benefit of AMF on cash crop performances in cover crop rotational systems.

Introduction

Growing cover crops can be an effective technique in crop rotations to enhance soil health and suppress weed populations (Snapp et al., 2005; Clark, 2008). Cover crops can cover fallow periods between cash crops that may be vulnerable to weed establishment or erosion. Short-season cover crops are also utilized and overwintered (live until spring) in temperate regions to reduce soil erosion, increase soil organic matter (Snapp et al., 2005; García-González et al., 2018), and enhance the biomass of plant growth-promoting microorganisms, such as phosphate solubilizing fungi and arbuscular mycorrhizal fungi (AMF) (Lehman et al., 2012; Higo et al., 2013; Karasawa & Takahashi, 2015; García-González et al., 2016).

It is well known that AMF can improve host plant phosphorus (P) uptake and growth performance (Smith & Read, 2008). These nutritional benefits of AMF can be enhanced through appropriate agricultural management (Gosling et al., 2006; Kahiluoto, Ketoja & Vestberg, 2012; Ryan & Graham, 2018). Indeed, many previous studies have indicated that AMF benefits agricultural crops, such as maize (Zea mays) and soybean (Glycine max), by increasing mycorrhization and AMF hyphal abundance in soil during early growth to enhance P uptake as well as maize and soybean yield (Gavito & Miller, 1998; Isobe et al., 2014). Additionally, agricultural practices may positively or negatively impact AMF taxonomic and functional community composition (Ohsowski et al., 2014; Xiang et al., 2016; Cofré et al., 2017; Xu et al., 2017; Zhao et al., 2017; Berruti, Bianciotto & Lumini, 2018; García de León et al., 2018; Higo et al., 2018b). Indeed, intensive agricultural practices, such as tillage, mono-cropping, seasonal fallow periods, and inorganic nutrient application, reduce both AMF populations and AMF benefits for field crops (Lehman et al., 2012; Higo et al., 2018a). On the contrary, adding cover crops during the winter period is also an effective technique in increasing indigenous AMF abundance in soil for following crops (Higo et al., 2010, 2015a). Thus, the introduction of cover crops, such as ryegrass (Lolium multiflorum), wheat (Triticum aestivum), mustards and oilseed rape (Brassicaceae) or leguminous crops, including hairy vetch (Vicia villosa Roth.) and clovers (Trifolium), in crop rotations in temperate agricultural systems is essential to reduce seasonal fallow and thus provides many benefits for subsequent crops and soil fertility (Karasawa & Takahashi, 2015; Higo et al., 2018b). Given these facts regarding AMF in cover crop rotational systems, it is important to understand which agricultural management, such as cover crop species and crop rotation, positively or negatively impacts individual AMF taxa and community structures in roots and soil.

However, what remains poorly understood and unclear is which factor, such as cover cropping or host crop identity, is driving increases in cash crop performance and AMF diversity. Little is also known about how cover cropping and host identity are linked to the diversity of AMF community structure, and the effectiveness of AMF in cover crop rotational systems to improve the robustness and reliability of agricultural management. Additionally, the best combination of cover crops and subsequent cash crops (maize and soybean) in rotations to increase the diversity or to alter the structure of AMF communities remains unknown. Thus, the goal of this research was to determine 1) which cover cropping or host identity is the most important factor in shaping AMF community structure in subsequent crop roots, and 2) how cover cropping and host identity alter the belowground communities of AMF in two different types of subsequent maize and soybean crops.

Materials and methods

Experimental design

A field experiment of cover crop rotation was established at Nihon University, in Kanagawa, Japan (35˚22’N 139˚27’E). The soil at the field site is classified as a volcanic ash soil (Allophonic andosol). In this study, we examined the impact of cover cropping and host crop type on the diversity of AMF communities colonizing roots of subsequent crops. We established cover cropping treatments in rotation with maize (Zea mays L., cv: Snow Dent 125 Wakaba) or soybean (Glycine max (L.) Merr., cv: Enrei). Three cover crop treatments including Italian ryegrass (Lolium multiflorum Lam., cv: Akatsuki), hairy vetch (Vicia villosa Roth., cv: Mamekko), and brown mustard (Brassica juncea (L.) Czern & Coss., cv: Karajin) or bare fallow was used to cover the soil surface during winter and the fallow period in annual cropping systems in a temperate region of Japan. There were three replicate plots per treatment arranged in a randomized complete block design. Each plot had an area of 9 m2 (3 × 3 m). The Italian ryegrass, hairy vetch, and brown mustard were sown in rows, spaced 30 cm apart, in the cropped treatment on November 9, 2016. The ryegrass seeds were sown at 400 kg ha−1 (because the rate of emergence in our research field was low) with N (ammonium sulfate) and K2O (potassium chloride) application rates of 100 kg ha−1. The hairy vetch seeds were sown at 80 kg ha−1 with N (ammonium sulfate) and K2O (potassium chloride) application rates of 50 and 100 kg ha−1, respectively. Brown mustard seeds were sown at 80 kg ha−1 with N and K2O application rates of 100 kg ha−1. The tops of the cover crops were cut close to the ground and removed on April 24, 2017. For fallow, weeds were manually removed during the winter period. Soil biochemical properties of the field after cultivation of cover crops are shown in Table S1.

We used a split-plot design to divide the 3 × 3 m of the cover crop experimental plots into 3.0 × 1.5 m plots for the two subsequent cropping plots (maize and soybean) (Fig. S1). Then, both maize and soybean cropping plots were replicated three times in 3 × 3 m plots. The maize and soybean seeds were sown at a spacing of 60 × 15 cm on May 29, 2017. In soybeans, the N and K2O application rates were 50 and 100 kg ha−1, respectively. In maize, the N and K2O application rates were 150 and 100 kg ha−1, respectively. No P fertilizer was applied in both maize and soybean cropping in this study.

Soil and root sampling and root staining

The soil samples were randomly taken from ten points using a soil core sampler (0−20 cm depth, four cm diameter) in each replicate and pooled to one composite sample on April 24, 2017. Maize and soybean root samples were taken at the 6 weeks after sowing on July 10, 2017. The root samples in both maize and soybean were collected from nine plants (to a depth of 15 cm, the diameter of 20 cm) per replicate. The root samples were maintained at −80 °C for DNA extraction and measurement of AMF colonization. We stained the root samples with a 5% (w/v) black ink-vinegar solution (Vierheilig et al., 1998). The AMF root colonization in the maize and soybean was measured according to Giovannetti & Mosse (1980).

Measurement of the aboveground plant biomass of maize and soybean

The aboveground plant parts of eight plants in soybean and maize were cut close to the ground at the 6 weeks after sowing and were randomly sampled on July 10, 2017. The aboveground maize and soybean plant biomass were measured in all plots. The aboveground plant biomass by maize and soybeans were determined after the samples were oven dried at 80 °C for 48 h.

DNA extraction from root samples

Genomic DNA was extracted from 100 mg of fresh root samples using DNA suisui-P kit (RIZO, Tsukuba, Japan) according to the manufacturer’s instructions. Briefly, root samples were soaked directly in liquid nitrogen, and then the root samples were crushed using a beads smasher MS-100 (TOMY, Tokyo, Japan). After crushing the roots, 360 μl of DNA suisui-P lysis solution and 40 μl of prepared additives with the DNA extraction kit were put into 1.5 ml tubes. Up to 400 μl of phenol: chloroform (1:1, v/v) was added to the tubes and mixed vigorously using a vortex mixer (Vortex-Genie 2, Scientific Industries, Tokyo, Japan). The samples were centrifuged at 15,000 rpm for 10 min, at room temperature. After the centrifugation, 200 μl of supernatant was transferred to a clean tube (1.5 ml) and 200 μl (equal volume) of 2-propanol was added. After that, the samples were centrifuged at 15,000 rpm for 10 min, at room temperature. After this centrifugation, the supernatant was discarded and added 800 μl of 70% ethanol into the 1.5 ml tube, and then this was centrifuged at 15,000 rpm for 10 min, at 4 °C. The supernatant was discarded and the remaining pellet was dried. We added 50 μl of Tris-EDTA (TE) buffer (1 mM Tris (pH 8.0), 0.1 mM EDTA (pH 8.0)) to dissolve the pellet, and this solution served as template DNA for polymerase chain reaction (PCR). The genomic DNA pellet was stored at −30 °C until use in the PCR.

Polymerase chain reaction amplification

The fragments in the fungal small subunit ribosomal DNA (SSU rDNA) were amplified using the first-round PCR. The universal primer NS31 (Simon, Lalonde & Bruns, 1992) and the fungus-specific primer AM1 (Helgason et al., 1998) were used in the first PCR round to amplify the 5’ end of the SSU rDNA region for Glomeromycotina. To decrease variations in the PCR process, samples were amplified in triplicate (Polz & Cavanaugh, 1998) using the fusion primer set in a PCR at 10 μl per subsample. The PCR was performed in 10 μl reaction mixtures with each containing 2× of reaction buffer, 0.4 μM of forward and reverse primers (10 μM), 1 U of Taq DNA polymerase (KOD multi & Epi, Toyobo, Japan), and 1 µl of template DNA using a Mastercycler ep Gradient (Eppendorf, Hamburg, Germany). The first-round PCR was performed according to the PCR cycle of Liang et al. (2008). The first PCR products were diluted 10-fold and used as templates for the second-round PCR using the primers AMV4.5NF and AMDGR (Lumini et al., 2010). The second-round PCR was performed in 10 μl reaction mixtures that each contained 2× of reaction buffer, 0.3 μM of forward and reverse primers (10 μM), 1 U of Taq DNA polymerase (KOD multi & Epi, Toyobo, Japan), and 1 µl of template DNA. The second-round PCR protocol was composed of an initial treatment at 94 °C for 2 min; 45 cycles of treatments at 98 °C for 10 s and at 60 °C for 10 s. Gel electrophoresis separated amplicons on 1% agarose gel, and the approximately 300 bp DNA amplicons were visualized by staining with ethidium bromide.

Illumina MiSeq sequencing and molecular diversity of AMF communities in roots

Three independent PCR products were pooled together and purified using NucleoSpin Gel and PCR Clean-up kit (Macherey-Nagel, Duren, Germany) to reduce potential early-round PCR errors, and quantified using UV spectrophotometry (DS-11 NanoPad, DeNovix Inc., Wilmington, DE, USA). The purified PCR amplicons were normalized before an Illumina MiSeq platform sequencing. The amplicons were paired-end (PE) sequenced on the Illumina MiSeq platform (Bioengineering Lab Co., Ltd, Kanagawa, Japan). In total, 24 sequencing libraries were constructed and independently sequenced. Sequence read processing was performed using QIIME version 1.9.1 (Caporaso et al., 2010).

The reads were truncated at any site that received an average quality score <20 over a 40 bp sliding window, and the truncated reads shorter than 40 bp were discarded using FASTX-Toolkit. Then, PE reads were assembled according to their overlap sequence with a minimum overlap length of 10 bp, while reads that could not be assembled were discarded. The clean sequences were analyzed using the FLASH (Fast length adjustment of short reads). Chimeric sequences were identified and removed using UCHIME in USEARCH. Operational taxonomic unit (OTU) grouping was performed using Silva and NCBI Genbank at 97% similarity. The representative sequences were checked against NCBI GenBank. The raw sequence data are available in the DNA Data Bank of Japan (DDBJ) (DDBJ Sequence Read Archive: DRA007103). Additionally, we performed a rarefaction analysis using the vegan package in R 3.5.0.

Statistical analysis

A generalized linear model was used to determine the effects of host identity and cover cropping on each parameter in this study in the multcomp package in R 3.5.0 (Horthorn, Bretz & Westall, 2008). Data for the significance of differences between subsequent host crop types among cover crop treatments were assessed using Student’s t-test. Differences among means were assessed using Tukey’s test (P-values < 0.05) using the multcomp package in R 3.5.0. In addition, the amplicons are relatively short (<400 bp), the fungal community that was obtained after pyrosequencing, was highly a good reflection of the fungal composition in the sample (Ihrmark et al., 2012). Moreover, Hill numbers (or the effective number of species) have been increasingly used to quantify the species/taxonomic diversity of an assemblage because they represent an intuitive and statistically rigorous alternative to other diversity indices (Chao et al., 2014; Hsieh, Ma & Chao, 2016). Thus, diversity of AMF OTU communities were measured based on the first three Hill numbers, such as species richness, Shannon diversity (the exponential of Shannon entropy), and Simpson diversity (the inverse Simpson concentration) using the “renyi” function the vegan package in R 3.5.0.

Additionally, the differences in the AMF community structures among host crop type and cover cropping was also examined by a distance-based redundancy analysis (db-RDA) in the vegan package in R 3.5.0 to analyze the relationship of cover crops and subsequent host crop type with respect to AMF community structures. The environmental variable of cover cropping and host crop type was coded as a dummy variable (0 and 1). Goodness-of-fit statistics (R2) of measured factors fitted to the db-RDA ordination of the AMF community structures were calculated using the “envfit” function in the vegan package with P-values based on 999 permutations (Oksanen et al., 2018). Differences in the AMF community structures between maize and soybean were also determined by non-metric multidimensional scaling (NMDS) with the “metaMDS” function. A permutational multivariate analysis of variance (PERMANOVA) was performed with 999 permutations using the “Adonis” function in the vegan package in R 3.5.0 to investigate if AMF community structures differed significantly between cover crops and subsequent host crop type. The “Bray-Curtis” metric of AMF community structure was used in the PERMANOVA and NMDS analysis (Chase et al., 2011).

Furthermore, we performed an analysis based on the multinomial species classification method (CLAM test) with the “clamtest” function of the vegan package in R 3.5.0 (Chazdon et al., 2011) to understand AMF OTUs showing a preference for maize or soybean. The CLAM test uses a multinomial model based on estimated species relative abundance in two habitat or host species (maize and soybean). The multinomial model implemented in the test minimizes biases due to different sampling intensities between the two habitats or host species being compared. The method permits a robust statistical classification of habitat specialists and generalists, without excluding rare species a priori (Chazdon et al., 2011). In this study, the model classifies species into one of four groups: (1) generalists; (2) maize specialists; (3) soybean specialists; and (4) too rare species.

In the diversity indices, db-RDA, PERMANOVA, NMDS, and CLAM test, we rarefied our dataset down to the lowest OTU abundance to compare evenly between samples regardless of sequencing depth (i.e. samples with greater sequencing depth can allow for greater detection of low-abundance taxa compared with more shallowly sequenced samples).

Results

Arbuscular mycorrhizal fungi (AMF) colonization and plant growth in maize and soybean roots

Overall, our results showed that cover cropping impacted AMF colonization in maize roots, whereas differences in AMF colonization of soybean were not impacted by cover cropping (Fig. 1). We found that AMF colonization of maize under Italian ryegrass and hairy vetch treatments was significantly higher than that of brown mustard and bare fallow treatments. In soybean, a similar tendency in AMF colonization with regard to maize cropping was observed. However, no significant differences in AMF colonization of soybean were found among cover cropping treatments.

Figure 1 Boxplots illustrating differences in group averages in root colonization of arbuscular mycorrhizal fungi (AMF) in maize and soybean at 6 weeks after sowing.

Bold horizontal lines represent median values; box margins ± SE and vertical lines represent minimum and maximum values of the groups. 1) Different letters in the maize or soybean plots among the cover crop treatments show a significant difference according to Tukey’s test at the 5% level. 2) R.M, V.M, M.M and F.M show the AMF root colonization of maize after cultivation of Italian ryegrass, hairy vetch, brown mustard cropping or bare fallow, respectively. R.S, V.S, M.S and F.S show the AMF root colonization of soybean after cultivation of Italian ryegrass, hairy vetch, brown mustard cropping or bare fallow, respectively.

According to Fig. 2, the aboveground plant biomass in maize did not vary among cover crop treatments in this study (Fig. 2A). The aboveground plant biomass under Italian ryegrass and brown mustard treatments tended to be higher than that of hairy vetch and bare fallow treatments. On the contrary, the aboveground plant biomass of soybean in the Italian ryegrass plot was more than double than those of hairy vetch and bare fallow treatments (Fig. 2B). Additionally, the aboveground plant biomass of soybeans in the brown mustard treatment was higher than those of hairy vetch and bare fallow treatments, as the brown mustard is a non-host crop.

Figure 2 Boxplots illustrating differences in group averages in aboveground plant biomass of maize (A) and soybean (B) at 6 weeks after sowing.

Bold horizontal lines represent median values; box margins ± SE and vertical lines represent minimum and maximum values of the groups. 1) Different letters in the maize or soybean plot among the cover crop treatments show a significant difference according to Tukey’s test at the 5% level. 2) R.M, V.M, M.M and F.M show the aboveground plant biomass of maize after cultivation of Italian ryegrass, hairy vetch, brown mustard cropping or bare fallow, respectively. R.S, V.S, M.S and F.S show the aboveground plant biomass of soybean after cultivation of Italian ryegrass, hairy vetch, brown mustard cropping or bare fallow, respectively. (A) Maize, (B) Soybean.

General sequencing information and taxonomic richness

In this study, a total of 1,087,375 paired-end sequences were obtained from the 24 libraries using the AMV4.5NF/AMDGR primer set. Of these, 873,458 sequences belonged to Glomeromycotina (corresponding to 80.3% of the total). We showed the OTU distributions of the obtained sequences in root samples (Table S2). In total, 83 OTUs were found in the AMF communities based on 97% similarity after the rarefaction process (Fig. 3). The AMF OTUs were also classified into one of four AMF families including Glomeraceae (average relative abundance: 60.7%), Gigasporaceae (22.0%), Acaulosporaceae (16.2%) and uncultured Glomeromycotina in roots (1.1%).

Figure 3 Rarefaction curves showing the amplicon sequencing depths in the maize and soybean root samples in this study.

The dashed lines show a minimum (25,116 sequences), mean (36,394 sequences) and median (36,857 sequences) number of obtained sequences in this study. The operational taxonomic units (OTUs) were compared across samples when samples were rarefied at 25,116 sequences.

Additionally, the OTU richness in maize after cultivation of brown mustard or bare fallow was higher compared with that from roots after cultivation of Italian ryegrass and hairy vetch, but this was not a significant difference (Fig. 4A). On the contrary, the Shannon (H’) and Simpson (1/D) indices in hairy vetch were significantly lower than those of other cover cropping treatments (Figs. 4B and 4C). In terms of other cover cropping treatments, no significant differences in the Shannon and Simpson indices were found (Figs. 4B and 4C). However, in soybean after cultivation of brown mustard or bare fallow, the OTU richness and Shannon index was lower compared with those from roots after cultivation of Italian ryegrass and hairy vetch (Figs. 4A and 4B). Moreover, at each parameter between maize and soybean roots, no significant difference was found (Figs. 4A–4C).

Figure 4 Boxplots illustrating differences in group averages regarding (A) operational taxonomic unit (OTU) richness, (B) Shannon index (H’) and (C) Simpson index (1/D) in the roots of maize and soybean at 6 weeks after sowing.

Bold horizontal lines represent median values; box margins ± SE and vertical lines represent minimum and maximum values of the groups. 1) Different letters in the maize or soybean plot among the cover crop treatments show a significant difference according to Tukey’s test at the 5% level. 2) R.M, V.M, M.M and F.M show the OTU diversity of maize after Italian ryegrass, hairy vetch, brown mustard cropping or bare fallow, respectively. R.S, V.S, M.S and F.S show the OTU diversity of soybean after Italian ryegrass, hairy vetch, brown mustard cropping or bare fallow, respectively.

Structures of AMF communities in the two different subsequent crops

Our results showed that the relative abundance of each AMF OTU and family tended to be different among cover cropping treatments when we compared the AMF communities in maize with that in soybean (Figs. S2A and S2B). Moreover, we found that the structure of AMF communities in Italian ryegrass versus hairy vetch treatments, and brown mustard versus bare fallow treatments in maize tended to shape a similar community structure. In Italian ryegrass and hairy vetch treatments, Glomeraceae was predominant. On the contrary, Gigasporaceae and Acaulosporaceae were predominant in brown mustard and bare fallow treatments. In soybean, the structure of AMF communities in brown mustard versus bare fallow treatments also tended to shape a similar community structure (Fig. S2A). In hairy vetch, brown mustard and bare fallow treatments, Glomeraceae were predominant, on the contrary, Gigasporaceae and Acaulosporaceae were predominant only in Italian ryegrass treatment. Additionally, we found that Glomeraceae was detected at a high frequency in both maize (56.2%) and soybean roots (62.2%) regardless of cover cropping treatments (Fig. S2A). Gigasporaceae was more abundant in maize (25.0%) compared with in soybean (20.6%), and the relative abundance of Acaulosporaceae in the maize (17.7%) was larger than that in soybean (16.2%). Among the OTUs of Glomeraceae, the OTUs of uncultured Glomus (Accession No., LT217508; LT723917; LT217431) were detected at a much higher frequency in both maize (28.8%; 8.5%; 8.3%, respectively) and soybean roots (22.4%; 18.4%; 13.5%, respectively) (Table S4; Fig. S2B). In turn, the OTU of Rhizophagus irregularis (HF968834) was also detected at a higher frequency in both maize (5.6%) and soybean roots (3.4%). Among the OTUs of Gigasporaceae, the OTU of Cetraspora pellucida (KX879059) was highly detected at a much higher frequency in both maize (19.5%) and soybean roots (16.1%). Additionally, the OTU of uncultured Acaulospora (LN890608) was detected at a much higher frequency in both maize (13.5%) and soybean roots (12.5%) among the OTUs of Acaulosporaceae.

Host preference of AMF communities in the two different subsequent crops

The AMF OTUs in various taxonomic lineages were also classified as “generalists,” commonly found in both subsequent crop species according to the multinomial species classification method (CLAM) test (Table S3 and S4). The proportion of common generalists in roots in this experiment among Italian ryegrass, hairy vetch or brown mustard versus bare fallow was 77.1%, 68.8% and 18.8%, respectively (Figs. 5A–5C). The proportion of detected OTUs that occurred specifically in only ryegrass, hairy vetch or brown mustard was 8.4%, 12.5%, and 33.8%, respectively. Moreover, detected OTUs that occurred specifically in only bare fallow measured 4.8%, 12.5%, and 41.2%, respectively, when comparing with Italian ryegrass, hairy vetch or brown mustard treatments. Additionally, most of the OTUs in root samples were common between maize and soybean, and the proportion of common generalists in this experiment for maize and soybean roots was 79.5% (Fig. 6). Detected OTUs that occurred specifically in only maize and soybean were 9.6% and 10.8%, respectively.

Figure 5 Screening of generalists and specialists of arbuscular mycorrhizal fungal communities in root samples among different cover cropping treatments.

The operational taxonomic units (OTUs) of arbuscular mycorrhizal fungi (AMF) commonly detected from both Italian ryegrass, hairy vetch or brown mustard and bare fallow treatments (circle), and those preferentially found from Italian ryegrass, hairy vetch or brown mustard (square) or bare fallow treatments (diamond). Rare AMF OTUs (triangle) were classified according to a multinomial species classification method (CLAM) test. (A) Italian ryegrass versus bare fallow treatment, (B) hairy vetch versus bare fallow treatment, and (C) brown mustard versus bare fallow treatment.

Figure 6 Screening of generalists and specialists of arbuscular mycorrhizal fungal communities in root samples between maize and soybean plots.

The operational taxonomic units (OTUs) of arbuscular mycorrhizal fungi (AMF) commonly detected from both maize and soybean (circle) samples; those preferentially found from maize (square) or soybean (diamond) samples were classified according to a multinomial species classification method (CLAM) test.

Relationships between AMF communities of cover cropping and subsequent host crops

We used db-RDA to identify the relationships among AMF communities in maize and soybean roots with that in cover crop management (Figs. 7A and 7B). The db-RDA trends clearly showed that the cover cropping noticeably altered the AMF community structure in the maize and soybean roots. In maize, the ordination diagram indicates that Italian ryegrass (R2 = 0.787, P = 0.003), hairy vetch (R2 = 0.721, P = 0.01) and bare fallow (R2 = 0.492, P = 0.028) contributed significantly to the variation in AMF root communities (Fig. 7A). However, brown mustard (R2 = 0.064, P = 0.737) did not contribute to the variation in the AMF root communities. In soybean, the ordination diagram indicates that Italian ryegrass (R2 = 0.558, P = 0.046) and bare fallow (R2 = 0.533, P = 0.039) contributed significantly to the variation in the AMF root communities (Fig. 7B). However, hairy vetch (R2 = 0.410, P = 0.076) and brown mustard (R2 = 0.475, P = 0.055) did not contribute to the variation in the AMF root communities. A PERMANOVA was also carried out to examine the effect of cover cropping on the AMF root communities in maize and soybean. The PERMANOVA showed that cover cropping significantly affected the AMF root community structure (maize; F = 4.647, P < 0.05, soybean; F = 6.339, P < 0.001) (Figs. 7A and 7B).

Figure 7 Distance-based redundancy analysis (db-RDA) biplot showing the relationship among the AMF communities and cover cropping in maize (A) and soybean (B).

Solid lines indicate significant effects, and dashed lines indicate non-significant effects.

Additionally, we used NMDS to identify the differences in the AMF communities between maize and soybean (Fig. 8). The differentiation of AMF communities between the two subsequent host crop species was not statistically significant (host crop; F = 1.238, P > 0.05, cover cropping; F = 2.048, P > 0.05). However, we found that the interaction with host crop species and cover cropping on the differences in the AMF communities in the maize and soybean roots was significantly different (interaction; F = 8.565, P < 0.001).

Figure 8 Non-metric multidimensional scaling (NMDS) plot showing differences in the arbuscular mycorrhizal fungal communities between maize (closed) and soybean (open) root samples.

Circles in the NMDS plot are 95% confidence ellipses of maize (solid line) and soybean (dashed line).

Discussion

Impact of cover cropping on AMF root colonization

Previous studies have shown that the introduction of cover crops or fallow affects AMF colonization of subsequent cash crops (Isobe et al., 2014; Karasawa & Takahashi, 2015; Higo et al., 2018a). Indeed, in the current study, brown mustard and bare fallow treatments significantly affected the AMF colonization of subsequent maize; however, these treatments did not affect the AMF colonization of subsequent soybean (Fig. 1), which is a finding that is in partial agreement with the results of previous studies (Karasawa, Kasahara & Takebe, 2002; Karasawa & Takebe, 2012). Some previous findings also indicated that the introduction of Brassicaceae plants reduced AMF root colonization in subsequent crops during early growth stages compared with the introduction of host crops (Sorensen, Larsen & Jakobsen, 2005; Koide & Peoples, 2012). Because brown mustard (a member of Brassicaceae) does not form symbiotic relationships with AMF in its roots due to the release of antifungal compounds, such as mustard oils or isothiocyanates. Indeed, antifungal compounds, such as isothiocyanates, decompose in soil owing to microbial effects and responses to organic matter (Morra & Kirkegaard, 2002). Schreiner & Koide (1993) showed that extracts from Brassicaceae roots reduced AMF spore germination after 5 and 7 days. However, AMF spore germination recovered to normal conditions after about 14 days, thereby suggesting that isothiocyanates and mustard oils may have an antimicrobial effect on soil microorganisms, including AMF spores. In fact, the maize and soybean growth performance in the early growth stage, even after brown mustard treatment, did not reduce with decreasing AMF root colonization (Figs. 2A and 2B), although brown mustard is a non-host crop, in agreement with the results of Higo et al. (2018b). Thus, adding brown mustard or fallow in a rotation with maize and soybean may not necessarily have negative impacts on the performance of subsequent maize and soybean in early growth stages.

Distribution of AMF communities in maize and soybean roots

Our results showed that the representatives of Glomeraceae (including a genus of Glomus and Rhizophagus) were found to be the main family, although Gigasporaceae (including a genus of Cetraspora, Gigaspora, and Racocetra), Acaulosporaceae (including a genus of Acaulospora) and uncultured Glomeromycotina were found in both maize and soybean roots (Fig. S2). These results are similar to the findings of published studies confirming that the AMF taxa of Glomeraceae are the most abundant in the AMF communities of maize (Isobe et al., 2011; Wang, White & Li, 2017; Higo et al., 2018b) and soybean roots (Higo et al., 2014, 2018a; García de León et al., 2018; Faggioli et al., 2019). The AMF taxa of Glomeraceae are particularly predominant in roots and arable soils (Senés-Guerrero & Schüßler, 2016; Xiang et al., 2016; Xu et al., 2017; Zhao et al., 2017) because they are better adapted to disturbed environments than other families in addition to having high sporulation rates for rapid recovery (Oehl et al., 2003). Moreover, Glomeraceae colonize throughout fragments of hyphae or mycorrhizal roots, thereby forming a hyphal anastomosis (Giovannetti, Azzolini & Citernesi, 1999), and therefore possessing the ability to reconstruct a network after mechanical disruption. However, Gigasporaceae of AMF propagates throughout spore dispersal or infection from an intact hypha (Biermann & Linderman, 2006; Schalamuk & Cabello, 2010). Thus, these factors promote the survival and spread of Glomeraceae family members in agricultural ecosystems, and the occurrence of this phenomenon can also be the result of adaptation to agroecological environments.

Additionally, Clavel, Julliard & Devictor (2011) indicated that in general, taxa of generalists are thought to be related to taxa of specialists with undisturbed habitats and disturbed habitats. Johnson (1993) has reported that AMF have different niches and are well known to prefer inhabiting different soils. Host plant types and environmental filters may favor AMF taxa that grow better in soils (Dumbrell et al., 2010). In fact, Börstler et al. (2008) and Séry et al. (2018) suggested that a Glomeraceae such as the generalist, R. irregularis, has been observed in many different types of fields and can have a high tolerance for environmental filters such as tillage. On the contrary, Gottshall, Cooper & Emery (2017) reported that AMF taxa of Gigasporaceae were affected by agricultural management practices with a low level of soil disturbance. Faggioli et al. (2019) also suggested that abundance of Gigasporaceae can rely on no-tillage practice. Additionally, intensive agricultural practices may reduce the taxa richness and change the AMF community structures according to favoring certain AMF taxa and disfavoring other AMF taxa (Gottshall, Cooper & Emery, 2017). This phenomenon shows the shift in the occurrence of generalist and specialist AMF taxa with respect to a range of host plant species or habitat (Helgason et al., 2007; Oehl et al., 2010). Thus, the fluctuation in abundance of AMF taxa (generalists, specialists and rare taxa) as a result of cover cropping or host crops (maize and soybean) (Figs. 5 and 6; Fig. S2; Table S3 and S4) could link to the preference of host identity or inhabiting preferable agricultural practices such as cover cropping among AMF taxa in maize and soybean roots.

Additionally, there are few but increasing evidence of the spatial distribution of AMF communities in soil at small scales using molecular techniques. For example, Mummey & Rillig (2008) also indicated the difference in the spatial distribution of AMF communities at a small (1 × 1 m) scale in a temperate grassland. Wolfe et al. (2007) also found that the spatial variation of AMF communities to be within a 2 × 2 m scale. In this study, the 3 × 3 m scale of our research plots may be small for performing experiments to determine the distribution of AMF communities in agricultural practices. However, we considered that the spatial distribution of specific AMF communities of our study responded to the presence of the crop in our crop rotational system, in accordance with the findings of previous studies (Wolfe et al., 2007; Mummey & Rillig, 2008). Taken together, our research plot scale is complete enough to compare the diversity of AMF communities with the impact of cover cropping and host identity.

Impact of cover cropping on the diversity of AMF communities

Although the benefits of a host plant may differ according to individual AMF taxa, there are little data with respect to how the diversity of AMF communities varies with cover cropping practices. To date, Higo et al. (2014) have demonstrated that the diversity of AMF communities colonizing subsequent soybean roots in a cover crop rotations system is clearly changed by rotation year, suggesting that climate or other environmental filters are more important than winter cover cropping. Furthermore, Higo et al. (2018b) and Turrini et al. (2016) showed that a difference in the structure of AMF communities colonizing subsequent maize roots in a cover crop rotation is independent of the identity of preceding crops. However, these studies used separate plots for the cover crop rotation experiment to assess the effect of cover crops on the AMF communities in maize or soybean roots. Thus, we examined three cover crop treatments (Italian ryegrass, hairy vetch, and brown mustard) and bare fallow for the diversity of AMF colonizing both subsequent maize and soybean roots. We found that hairy vetch before planting maize had a negative impact on the diversity of AMF in roots of maize, but not for that of soybean (Figs. 4B and 4C). In soybeans, brown mustard before planting soybean had a negative impact on the diversity of AMF in roots. Additionally, the shift of AMF communities in both maize and soybean was obvious from the results of db-RDA, which showed how cover cropping changed the AMF community structure in both subsequent crop roots (Figs. 6A and 6B), in agreement with the observations of a previous study (Higo et al., 2018a, 2018b). These findings indicate that cover crop species may induce a shift in the diversity and structure of AMF communities in soils, and the differences in the soil AMF communities after the different types of cover cropping may change the diversity of AMF communities in subsequent crops.

Impact of host identity on the diversity of AMF communities

Several studies have reported that host preference between host plants and AMF taxa may impact AMF community structure (Helgason et al., 2002; Yang et al., 2012). We expected that if host identity is a strong factor in determining AMF communities compared with cover cropping in the split-plot of our experiment, the AMF community structure associated with maize and soybean roots after cover cropping would be distinct host AMF community structures between maize and soybean roots. This would hold true even if different cover crop treatments were to shape a specific assemblage of AMF taxa in soils before planting maize and soybean in our cover crop rotation. However, our NMDS and PERMANOVA result demonstrated that host identity did not change the AMF community structure of maize and soybean roots (Fig. 8). Furthermore, host identity did not impact the richness, diversity, or structure of the AMF community in the roots of maize and soybean at the early growth stage, and this finding was somewhat unexpected. Gosling et al. (2013) reported that soil P concentration had a greater impact on host identity. On the contrary, abiotic environmental filters, such as land-use type (Bainard et al., 2014), growing season (Higo et al., 2015b), and soil P (Bainard et al., 2014), appear to override the impact of host identity if their impacts are large enough compared with host identity.

Additionally, Isobe et al. (2011) have shown that the AMF community structure in soybeans grown in cool and temperate regions are drastically different, thus suggesting climate condition, such as growth temperature, can also be a strong determinant in shaping AMF community structures. To date, only a few studies have also shown that land-use type and host identity have interactive effects on root AMF community structure in grasslands (Vályi, Rillig & Hempel, 2015; Ciccolini et al., 2016) and agricultural fields (Higo et al., 2015b, 2016; Ciccolini et al., 2016), whereas interactions between land use and other factors, such as soil biochemical properties, have also been reported in temperate areas (Jansa et al., 2014; Ciccolini, Bonari & Pellegrino, 2015). Indeed, a specific reason for why there was a significant interaction in the AMF communities between maize and soybean, was uncertain in this study (Fig. 8). Thus, we hypothesize that one possible explanation for the consistency in AMF communities between both subsequent crops may be that the impacts of specific root exudate patterns in both maize and soybean on the host selectivity of AMF taxa may be weakened due to planting maize and soybean within the same plot, or may be weakened at a too early growth stage, as suggested by Higo et al. (2015b) and Borrell et al. (2017). However, how and whether the selectivity of the host plant impacts the pattern of AMF communities in agricultural systems, remains unknown. Thus, further investigation into the relationships among a pattern of AMF communities would be required to better understand whether and how host selectively impacts the AMF communities in cover crop rotational systems.

Conclusions

In conclusion, cover cropping can be an important factor for shaping AMF communities in maize and soybean roots rather than their host identity in a cover crop rotation. In addition, different cover crop species may have different influences on the diversity of AMF community structure in both maize and soybean roots. Additionally, interactions of cover cropping and host identity may partially relate to shaping the diversity and structure of AMF communities in maize and soybean roots in this study. These differences in the AMF community structure may relate to maize and soybean production in cover crop rotational systems. However, we still need to clarify whether and how host identity interacts with the diversity and structure of AMF communities associated with maize and soybean in cover cropping systems. This knowledge will give useful information on appropriate cover crop choices in cover crop rotational systems.

Supplemental Information

Supplemental Information 1 Summary of the experimental design of the cover crop rotational study.

Click here for additional data file.

Supplemental Information 2 Relative abundance of arbuscular mycorrhizal fungi (AMF) in maize and soybean roots at 6 weeks after sowing.

1) R.M, V.M, M.M and F.M show the relative abundance of OTU in maize after cultivation of Italian ryegrass, hairy vetch, brown mustard cropping or bare fallow, respectively. R.S, V.S, M.S and F.S show the relative abundance of OTU in soybean after cultivation of Italian ryegrass, hairy vetch, brown mustard cropping or bare fallow, respectively. A = Family-based abundance, B = OTU-based abundance.

Click here for additional data file.

Supplemental Information 3 Soil biochemical properties after cultivation of winter cover crops or bare fallow.

Click here for additional data file.

Supplemental Information 4 Description of detected OTUs of AMF in maize and soybean root samples according to an Illumina Miseq Platform.

Click here for additional data file.

Supplemental Information 5 Generalist or specialist found from root samples among cover cropping treatments.

Click here for additional data file.

Supplemental Information 6 Generalist or specialist found from both maize and soybean root samples.

Click here for additional data file.

Additional Information and Declarations

Competing Interests

Author Contributions

DNA Deposition

Data Availability

The authors declare that they have no competing interests.

Masao Higo conceived and designed the experiments, performed the experiments, analyzed the data, contributed reagents/materials/analysis tools, prepared figures and/or tables, authored or reviewed drafts of the paper, approved the final draft.

Yuya Tatewaki performed the experiments, contributed reagents/materials/analysis tools, guidance on lab techniques.

Kento Gunji performed the experiments, contributed reagents/materials/analysis tools, guidance on lab techniques.

Akari Kaseda performed the experiments, contributed reagents/materials/analysis tools.

Katsunori Isobe contributed reagents/materials/analysis tools, guidance on lab techniques.

The following information was supplied regarding the deposition of DNA sequences:

The raw sequence data are available in the DNA Data Bank of Japan (DDBJ) (DDBJ Sequence Read Archive: DRA007103).

The following information was supplied regarding data availability:

The raw sequence data are available in DNA Data Bank of Japan (DDBJ) Sequence Read Archive at DRA007103/Bio Project ID: PRJDB7241.

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
