# Peer review of "Cover cropping can be a stronger determinant than host crop identity for arbuscular mycorrhizal fungal communities colonizing maize and soybean"

_PeerJ, doi:10.7717/peerj.6403_

## Round 0.1 · original submission · Major Revisions

Both reviewers and myself found your work interesting and providing new findings. However there are many sections and some issues in the manuscript that should be addressed to make the manuscript clearer and the results more supported. Please I suggest to take into consideration and address all comments made by the reviewers in order to improve your manuscript.

Reviewer 1 ·

Basic reporting

Please, find my comments on general comments for the author.

Experimental design

Please, find my comments on general comments for the author.

Validity of the findings

Please, find my comments on general comments for the author.

Additional comments

L28 How do you define generalist?
L54 Please, consider a citation to:
RYAN, M. H. & GRAHAM, J. H. (2018). Little evidence that farmers should consider abundance or diversity of arbuscular mycorrhizal fungi when managing crops. New Phytologist early view.
L56 I missed a mention to the changes that agricultural practices may cause on AM fungal taxonomic and functional community composition. Please, consider a citation to:
GARCÍA DE LEÓN, D. CANTERO, J. J. MOORA, M. ÖPIK, M. DAVISON, J. VASAR, M. JAIRUS, T. & ZOBEL, M. (2018). Soybean cultivation supports a diverse arbuscular mycorrhizal fungal community in central Argentina. Applied Soil Ecology 124, 289-297.
OHSOWSKI, B. M. ZAITSOFF, P. D. ÖPIK, M. & HART, M. M. (2014). Where the wild things are: looking for uncultured Glomeromycota. New Phytologist 204(1), 171-179.
L60-L61 Sentence „Thus […] (Njeru et al. 2014)“ does not seem a consequence of previous sentece. Please, consider rephrasing.
L74-LL88 largely repeats the information of previous paragraphs.
L80 Is Noelia Cofré et al 2017 cited correctly? Is Noelia the first name or a surname?
L91 What level is „host identify“ define at? Is it cultivar, plant species, groups of crops? For instance, should a manuscript regarding wheat focused on wheat „wheat cultivar Arabella 2012“, „wheat“ or „cereals“? How it this for maize and soybean in relation to AM fungi?
L94 What do you mean by „improve the diversity or structure of AM fungal communities“? Improvement implies a goal (i.e. AM fungal community of reference) to achieve? Is this true for Integrated Mycorrhizal Technology? Should the production of food for a growing human population the solely target of agriculture? I personally do not think that agriculture has one unique aim, but some readers may believe food prodution is the only priority for agriculture. As far as I know there are not any „AM fungal community of reference“, which may represent a goal for farmer or scientistist to achieve. If the authors knew of any, I would extremely thankful for them to cite it. If there aren’t, a clarification is needed.
L102-L120 Experiment seems well design.
L117 Perhaps, the repetition of „100 and 100“ is not needed.
L121-L127 Plot scale seems rather small. That does not preclude the interest of study. However, many readers will appreciate a paragragh in the discussion dedicated to how your results might me scale-up.
L147 Could you please summarize the manufacturer’s instructions? Readers from other regions might not be familiar with the kits produced by RIZO, Tsukuba, Japan. For instance, in Europe and North America the DNA extraction market for AM fungi-related purposes is largely dominated by MO BIO Laboratory Inc., Carlsbad, CA, USA.
L150-L168 Nested PCR methods seems sound to me.
L187-L189 Does rarefaction imply that abundance data were used? What would be the effect on the results of using exclusively presence/absence data? There is an open debate that discourage the use of molecular data to estimate the abundance of AM fungi. Please, consider incorporating –and change the methods accordingly, if neccesary- the following citation in the manuscript:
Gamper HA, Young JPW, Jones DL, Hodge A. 2008. Real‐time PCR and microscopy: are the two methods measuring the same unit of arbuscular mycorrhizal fungal abundance?. Fungal Genetics and Biology 45: 581–596.
L192 Perhaps, it is my personally opinion. There are several statistical schools, but I do not particularly like transforming data for fitting a normal distribution. Instead, there are plenty of other statistical tools such as generalized linear models. Could you please explore –and show it to me- how robust your results are to the chosen methodology.
L196-L199 Please, revisit comment on L187-L189.
L200-L215 I agree the methodology described in this paragraph is generally finne. However, I have several concerns: (a) The significance of explanatory variables with function „adonis“ depends on the order they are included in the model in. Could you try and reproduce your results with function „Adonis“ [note capital A], which is robust to this problem?; (b) Rarefaction is a tricky issue. I personally feel rarefaction to the minimum number of sequences waste a lot of information. Perhaps, rarefaction to the median number of sequences might be a good solution. Plus, some readers will appreciate some information about the statistical distribution of the number of sequences (e.g. minimum, Q1, mean, median, Q3, standard deviation, standard error, maximum). It might be shown in the results, but this information is crucial to decide about the rarefaction method. Indeed, you used iNEXT package. iNEXT function from iNEXT package not only interpolates (i.e. rarefaction), but also extrapolates. I find hard to understand how the rarefaction to the miinimum and the use of Chao Extrapolation Index (i.e. INext; see Hsieh et al 2016, Journal of Vegetation Science)
See also my previous criticism about rarefaction on L187-L189 and L196-L199.
L216-L218 Unfortunately, I am not familiar with clamtest function. Many readers might be in the same situation as me. Could you briefly explain what it does?
Final comment: In general, I find the paper extremely interesting. However, it still contains some methodological issues that certainly require the authors’ attentions. Solving these issue could potentially affect some results. Therefore, I decide not comment on results and discussion at this time. I will be very happy to review this manuscript again, once the methodological flaws have been addressed. At the moment, I cannot trust the results are robust enough to comment on them because some of them might change.

Reviewer 2 ·

Basic reporting

Literature references are update and according to the topic studied in the manuscript. I have found that some of the references cited in the text are not included in the list of the references and vice versa. Please, check and correct the references in the text and the list of references.
For example, “Clark, 2007” (line 45), “Oehl et al, 2003” (line 355), “Giovannetti et al., 1999” are not in the list of references.
In line 434, is Borrell et al. (2017) or Borrell et al. (2016) as is written in the list of references?
Line 574, where is this reference cited in the text?

Experimental design

No comment.

Validity of the findings

No comment.

Additional comments

The manuscript titled “Is host crop identity a stronger determinant than cover cropping for arbuscular mycorrhizal fungal communities colonizing maize and soybean roots in a cover crop rotation?” contains novel results that help to understand the effect of the different crops species used in crop rotations on the community structure of arbuscular mycorrhizal fungi.
I think that the title could be reduced without repeating words as cover crops and cover cropping. It will make it clearer.

Introduction:
Some parts of the introduction could be condensed. Some points of the first paragraph are repeated in the third. For example, the information in lines 65-67 is similar to the line 48.

Materials and Methods:
Line 112, you wrote “temperate regions of Japan” and it seems that your experiment is located in different regions around Japan. Maybe you could write “in annual cropping systems in a temperate region of Japan”.
Line 113-119. Is ryegrass treatment included in these activities (sowing and fertilization)? You should add how you sowed the ryegrass and if it was fertilized or not.
Line 122. In Figure S1, there is a sowing cover crop date in the fallow treatment. This cell should be empty to avoid confusion since it seems that a cover crop was planted in these plots.
Line 133. Did you use an auger to take the samples? Please, specify the apparatus.

Results:
Lines 247-249. Figure 3A shows OTU richness and you are talking about number of OTUs in this sentence. Please, clarify this part.
Lines 263-265. If you are comparing the different cover crops treatments within each cash crop (maize or soybean), the Figure S3A could be clearer if you group the different treatments for maize to the left and the treatment for the soybean to the right as you did in Figure 1.
Line 302, you talk about results from 2014 and 2015 but your experiment started in 2016. Please, clarify this part.

Discussion:
Line 328-329. How did the mustard and fallow affect the AMF colonization? Because ryegrass and vetch treatments also affected the AMF colonization by increasing significantly the colonization compared with fallow and mustard treatments. Please, clarify this part including whether they reduce or increase the colonization compared with which treatment.

---

## Round 0.2 · Minor Revisions

There are a few comments made by reviewer 2 that needs clarification. You do not need to add the extra reference as suggested, if you do not think it is necessary.

Reviewer 1 ·

Basic reporting

This resubmission has satisfactorily addressed all my previous comments. I do not have any further comments.

Experimental design

See basic reporting.

Validity of the findings

See basic reporting.

Additional comments

See basic reporting.

Reviewer 2 ·

Basic reporting

Please, find my comments on general comments for the author.

Experimental design

Please, find my comments on general comments for the author.

Validity of the findings

Please, find my comments on general comments for the author.

Additional comments

Introduction:
Lines 51-52. The reference of “García-González et al, 2016” could be more appropriate in line 54. On the other hand, another example of organic matter improvement by cover cropping is García-González et al. 2018 (García-González, I., Hontoria, C., Gabriel, J. L., Alonso-Ayuso, M., & Quemada, M. (2018). Cover crops to mitigate soil degradation and enhance soil functionality in irrigated land. Geoderma, 322, 81-88).
I found large the second paragraph of the introduction. I think that lines from 62 to 66 could be reduce or rewrite with the information of the following sentences.
Line 81: It is not clear for me the new sentence “the level of host plant species in this study”. I do not know if you could use “host species identity”.
Results:
Lines 277-279. This part is confusing for me. Frist, you said there are treatments with higher or lower OUT richness and then in a second sentence you did not find statistical differences among treatments. It is similar in the following lines when you talk about Shannon index on brown mustard. Please, rewrite these sentences and make them clearer because my first impression was that there were statistical differences. In addition, you must check the reference of the number of figures. Figure 3B does not exist.
In lines 347-350 you explain that you performed a PERMANOVA to examine the relative importance of each cover cropping for the AMF communities. And where are these results? Where is the importance of each cover crop on AMF communities?

---

## Round 0.3 · accepted · Accept

Thanks for taking into consideration the last comments made by the reviewers. Your manuscript is now ready for publication.

#